# SD-KDE: Score-Debiased Kernel Density Estimation

**Elliot L. Epstein**[1]*    **Rajat Dwaraknath**[1]*    **Thanawat Sornwanee**[1]*

**John Winnicki**[1]*    **Jerry Weihong Liu**[1]*

[1]**Stanford University, Stanford, CA 94305, USA**
{epsteine, rajatvd, tsornwanee, winnicki, jwl50}@stanford.edu

## Abstract

We propose a novel method for density estimation that leverages an estimated score function to debias kernel density estimation (SD-KDE). In our approach, each data point is adjusted by taking a single step along the score function with a specific choice of step size, followed by standard KDE with a modified bandwidth. The step size and modified bandwidth are chosen to remove the leading order bias in the KDE, improving the asymptotic convergence rate. Our experiments on synthetic tasks in 1D, 2D and on MNIST, demonstrate that our proposed SD-KDE method significantly reduces the mean integrated squared error compared to the standard Silverman KDE, even with noisy estimates in the score function. These results underscore the potential of integrating score-based corrections into nonparametric density estimation.

## 1 Introduction

Kernel density estimation (KDE) (Rosenblatt, 1956; Parzen, 1962) is a widely used nonparametric method for estimating an unknown probability density function from a finite set of data points. The classical KDE effectively smooths the data by convolving with a kernel function, such as the Gaussian kernel, and then normalizing the result to obtain a density estimate. KDE finds application in diverse fields such as anomaly detection, clustering (Campello et al., 2013), data visualization (Scott, 2012), nonparametric statistical inference (Guerre et al., 2000; Zhang et al., 2008), and dynamical systems (Hang et al., 2018).

The classical KDE suffers from a well-known bias-variance trade-off, controlled by the choice of kernel bandwidth (Silverman, 1986). Larger bandwidths lead to smoother estimates with lower variance but higher bias, while smaller bandwidths yield more variable estimates with lower bias (Rosenblatt, 1956; Parzen, 1962). This trade-off is particularly damaging in cases with highly variable density functions, where the bias can dominate the estimation error.

Recent advances in score-based generative modeling and diffusion processes have demonstrated the power of using the score function—the gradient of the log-density—to reverse a forward process of noise injection, effectively reconstructing the underlying data distribution (Ho et al., 2020). Notably, methods such as score matching (Hyvärinen & Dayan, 2005) and its deep learning extensions, diffusion models, (Song & Ermon, 2019) provide robust estimates of the score function even in complex, high-dimensional settings, without requiring density estimation.

In this work, we investigate whether incorporating knowledge of the score function into the KDE framework allows us to push the Pareto frontier of the bias-variance trade-off. We propose a method

---

*Equal contribution.

39th Conference on Neural Information Processing Systems (NeurIPS 2025).

to debias the KDE using the score function to improve density estimation accuracy. Specifically, our method adjusts each data point by taking a small step in the direction of the estimated score, and then performs KDE with a modified bandwidth, as illustrated in Figure 1. Intuitively, taking a step along the score sharpens the sample distribution, which counteracts the smoothening effect of applying the KDE. We find that with a carefully chosen combination of step size and KDE bandwidth, we remove the leading order bias in the KDE, resulting in a more accurate, debiased density estimate. Crucially, SD-KDE also works with empirical scores obtained directly from a vanilla KDE (via the gradient of the log of the KDE density estimate); no learned diffusion model is required.

In summary, our contributions are the following:

1. We propose Algorithm 1, our method for score-debiased kernel density estimation (SD-KDE).

2. We provide asymptotically optimal bandwidth and step size selection for Algorithm 1 (Theorem 1), achieving the asymptotic mean integral square error (AMISE) of order $\mathcal{O}\left(n^{-8/(d+8)}\right)$, instead of the $\mathcal{O}\left(n^{-4/(d+4)}\right)$ achieved by a standard KDE (Silverman, 1986).

3. In Section 3, we numerically corroborate our theoretical results on 1D and 2D synthetic datasets and observe strong agreement with the asymptotic scaling identified in Theorem 1.

## 2  Method and Theoretical Results

---
**Algorithm 1** Score-Debiased Kernel Density Estimation
---
**Require:** Data $\{x_i\}_{i=1}^n$, score estimator $\hat{s}$, kernel $K$, KDE bandwidth $h$, score step size $\delta$
 1: Take a single step along the score function: $\widetilde{x}_i \leftarrow x_i + \delta\hat{s}(x_i)$ for $i = 1, \ldots, n$
 2: Compute the debiased kernel density estimate: $\hat{p}(x) = \frac{1}{nh^d} \sum_{i=1}^n K\left(\frac{x - \widetilde{x}_i}{h}\right)$
---

**Theorem 1** (Optimal Bandwidth and Step Size selection for $Algorithm$ 1)**.** *Let $\{x_i\}_{i=1}^n$ be i.i.d. samples from a smooth density $p$ in $\mathbb{R}^d$. Let $\hat{s}$ be the exact score function of $p$. Let $K$ be a symmetric kernel with mean $0$, covariance $\int uu^\top K(u)du = I$, and a convergent Taylor series. The debiased kernel density estimate $\hat{p}$ obtained by running $Algorithm$ 1 with bandwidth $h$ and step size $\delta$ is given by*

$$\hat{p}(x) = \frac{1}{nh^d} \sum_{i=1}^n K\left(\frac{x - (x_i + \delta\hat{s}(x_i))}{h}\right).$$

*The asymptotically optimal bandwidth and step size for $Algorithm$ 1 are given by*

$$h_{opt} = \mathcal{O}\left(n^{-1/(d+8)}\right), \quad \delta_{opt} = \frac{h_{opt}^2}{2}.$$

*The resulting debiased kernel density estimate $\hat{p}$ satisfies*

$$\mathrm{MISE} := \mathbb{E}\left[\int (\hat{p}(x) - p(x))^2 dx\right] = \mathcal{O}\left(n^{-8/(d+8)}\right).$$

We include the detailed proof of Theorem 1 in Section 4.

**Corollary 1.** *If the estimate score $\hat{s}(\cdot)$ is not equal to the actual score $s(\cdot)$, the bandwidth is $h$, and the stepsize is $\frac{h^2}{2}$ as in Theorem 1, then the bias is given by*

$$\mathbb{E}\left[\hat{p}(x) - p(x)\right] = -\frac{h^2}{2}\left[(\hat{s}(x) - s(x))\nabla p(x) + p(x)\nabla(\hat{s}(x) - s(x))\right] + \mathcal{O}(h^4).$$

The proof for this corollary directly follows the proof for Theorem 1.

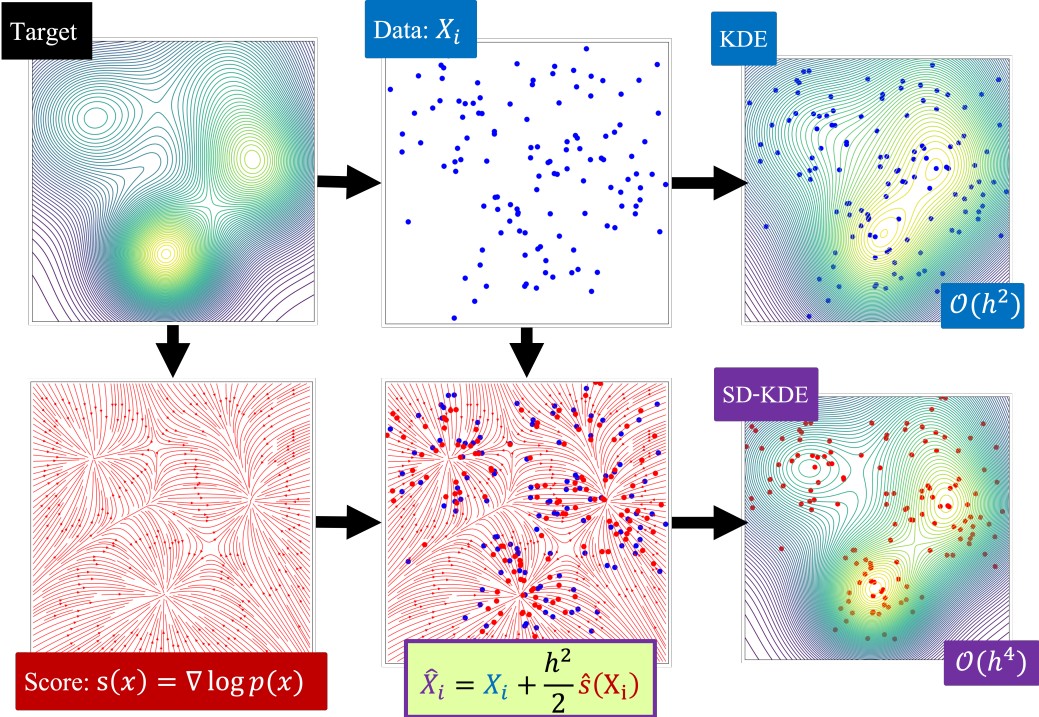

Figure 1: First schematic diagram on SD-KDE. The estimation objective is to estimate the target distribution pdf. In the conventional setting, we have finite samples from the distribution (blue box). Using only this information, we can perform KDE to estimate the probability density function. However, if we have access to the score function, we can combine the data points and score function to get SD-KDE. By fixing the kernel bandwidth to be $h$, we will get that the vanilla KDE and SD-KDE have a pointwise variance of order $\mathcal{O}\left(\frac{1}{nh^d}\right)$. However, SD-KDE reduces the pointwise bias from $\mathcal{O}(h^2)$ to $\mathcal{O}(h^4)$ per Theorem 1.

**Discussion.** Theorem 1 demonstrates that, when a score oracle is available, one can eliminate the asymptotically dominant term, thereby reducing the bias from the conventional order of $\mathcal{O}\left(h^4\right)$ to $\mathcal{O}\left(h^8\right)$. Although a higher-order kernel—such as the effective spline kernel described by Silverman (1984)—can similarly achieve a similar bias reduction, it typically introduces regions where the estimated density assumes negative values. This drawback poses a significant practical challenge, as the numerical normalization of the resulting probability density function is computationally intractable (Song & Ermon, 2019).

We note that our method flexibly allows a variety of kernels to be used for KDE, since the only requirement for the kernel is in symmetricity and covariance structure, both of which can be conveniently satisfied (Chen, 2017).

Although Theorem 1 requires the knowledge of the score function, we observe empirically that a small discrepancy of the estimated score function and the underlying score function under some level may only have minimal effect on the performance. See Section 3 for more details.

## 3 Experiments

### 3.1 1D Synthetics

**Experimental setup.** We test the empirical performance of the SD-KDE[1] method on density estimation of 1D Gaussian mixture models, and include a similar analysis for Laplace mixture models

---

[1]We open-source our implementation of SD-KDE at `https://github.com/Elliotepsteino/SD-KDE`.

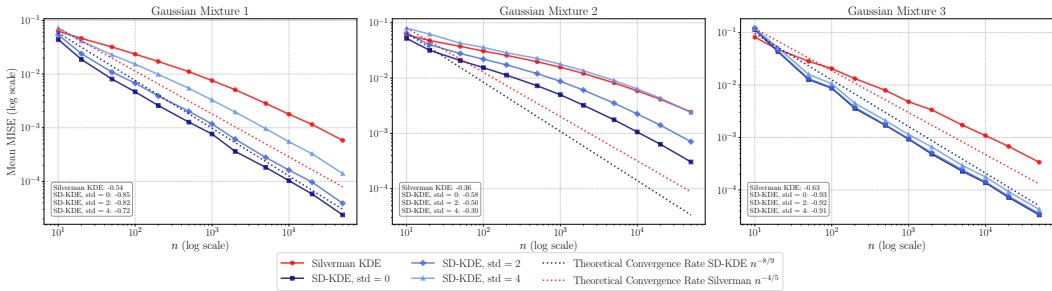

Figure 2: MISE error as a function of $n$ for the three Gaussian mixtures for Silverman vs. SD-KDE. Each point is the average MISE over 50 random seeds. The slopes inside each subplot are fitted regression lines in log–log scale, indicating how quickly each method's error decays as $n$ increases.

in Appendix A. We sample data from three mixtures, where

$$p(x) = \pi \mathcal{N}\big(x \mid \mu_1, \sigma_1^2\big) + \big(1 - \pi\big)\mathcal{N}\big(x \mid \mu_2, \sigma_2^2\big)$$

and each mixture's parameters $(\pi, \mu_1, \sigma_1, \mu_2, \sigma_2)$ are outlined in Table 1. We compare the SD-KDE method with a baseline based on the classical Silverman KDE, using Silverman's bandwidth formula (Silverman, 1986), given by $h = 0.9 \cdot \min(\hat{\sigma}, IQR/1.34) \cdot n^{-1/5}$, where IQR is the inter-quartile range. To investigate how sensitive our method is to the estimation accuracy of the score function, we test the performance of our method when only given access to a noisy score function estimate, e.g. we observe $\tilde{S}(x) = S(x) + \epsilon$, where $S(x)$ is the score function and $\epsilon \sim N(0, \sigma^2)$ for a given standard deviation $\sigma$. Performance is evaluated with *mean integrated squared error* (MISE).

Most of the experiments in the paper were conducted on a Linux cluster with 5 NVIDIA RTX A6000 GPUs, each with 49140 MB memory, running on CUDA Version 12.5. The cluster has 256 AMD EPYC 7763 64-Core Processor CPUs. Some experiments were also conducted on a MacBook Air (2022) equipped with an Apple M2 chip and 16 GB of unified memory. All experiments took less than 1 hour to run.

**SD-KDE is robust to noisy score function estimate.** In Figure 2, we show the MISE of the SD-KDE (as a function of the number of observed samples, $n$), with varying degree of added noise and compare to the Silverman KDE. Each point in the plot represents an average over 50 seeds. We see that the SD-KDE method has a significantly better asymptotic scaling than the Silverman baseline, up to a score function noise level with $\sigma = 4$. Even in the presence of a highly noisy score function, we find the SD-KDE method provides a significant gain. We also display the fitted regression slope associated with each line, along with the theoretical asympototic convergence rate of $n^{-8/9}$. We note the close tracking between the SD-KDE asymptotic decay ($-0.85$ for mixture 1, and $-0.93$ for mixture 3) compared with the theoretical predicted decay ($= -8/9 = -0.86$). For mixture 2, all models have weaker performance due to the challenging mixture shape, indicating that larger $n$ is needed to reach the theoretical decay rate. For $n = 5 \times 10^4$, the SD-KDE has an order of magnitude smaller MISE error on average across 50 seeds compared with the Silverman method.

**SD-KDE consistently beats Silverman baseline.** In Figure 3, we examine the consistency of the performance gains across multiple data seeds for $n = 100$. We observe that the SD-KDE method is consistently better than the Silverman baseline; for mixtures 1 and 2, SD-KDE method outperforms for all 100 samples, and for the third mixture, it is better in 95% of samples.

**SD-KDE with the Empirical Score** We now relax the assumption that the Score function is given, rather, we use Silverman KDE to approximate the score function, and then apply SD-KDE based on this estimated score. We call this method Emp-SD-KDE. Figure 4 shows that Emp-SD-KDE method greatly improves on the standard Silverman KDE, without any assumptions on knowing the true score function. Figure 5 shows how the empirical score is used.

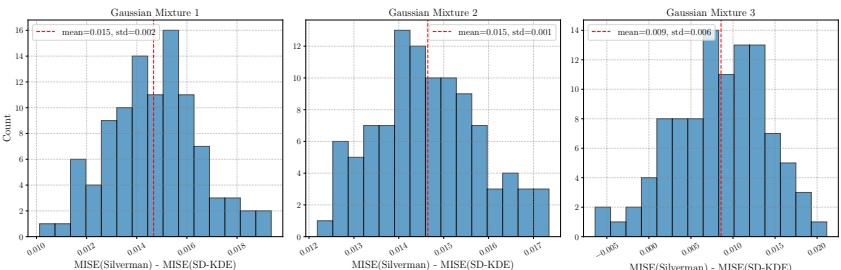

Figure 3: Histogram of MISE difference of the SD-KDE method and the Silverman method, for $n = 100$ samples and 50 random seeds per mixture. The SD-KDE method is consistently having lower MISE than the Silverman baseline; for mixtures 1 and 2, SD-KDE method outperforms for all 100 samples, and for the third mixture, it is better in 95% of samples.

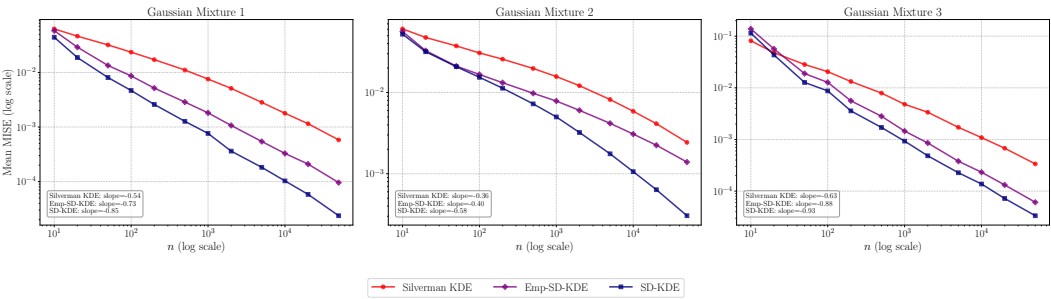

Figure 4: MISE error as a function of $n$ for each of the three gaussian mixtures. For each point, we compute the MISE with 50 random seeds per mixture. Each subplot plots the mean integrated squared error as a function of $n$. The legend compares Silverman KDE, Emp-SD-KDE (estimating the score from the data), and SD-KDE (ground truth score). The slopes inside each subplot are fitted regression lines in log–log scale indicating how quickly each method's error decays as $n$ increases.

## 3.2 2D Synthetics

We present preliminary results on 2D synthetic tasks, a spiral distribution (Figure 6) and a mixture of Gaussians (Figure 15), following Liu et al. (2020); Grathwohl et al. (2019).

In Figure 6, we compare the 2D Silverman method to SD-KDE for the spiral distribution. We compare the accuracy of our method using the true score function to using an estimate of the score function obtained by training a denoising diffusion probabilistic model (DDPM) from scratch on the training data. For the diffusion model architecture, we use a 3-layer MLP with hidden dimension 512, and we train the model with Adam for 1500 steps. We use 1000 diffusion steps during training. Using the true score function, our proposed method outperforms the Silverman method both qualitatively (via visual assessment) and quantitatively, as measured by the MISE. When employing the score estimated from the diffusion model, our method achieves performance comparable to that of the Silverman method. We attribute this discrepancy with the method under the true score parameter primarily to challenges encountered during the training of the diffusion model rather than to any inherent limitations of the method itself, particularly given the accuracy observed when using the true score.

## 3.3 Iterated SD-KDE: Incremental Improvements to KDE

In a 1D Gaussian mixture experiment, we examine an iterative application of SD-KDE to further improve the density estimate. We work with a gaussian mixture centered at $\pm 0.5$, each with standard deviation 0.2, 0.3 with weights 0.7 and 0.3. For this experiment, we will sample 1000 points and hold the bandwidth constant at 0.15. We start with a vanilla Gaussian-kernel KDE fit to the mixture data and compute its the closed form solution or approximation of its score, then apply SD-KDE (one

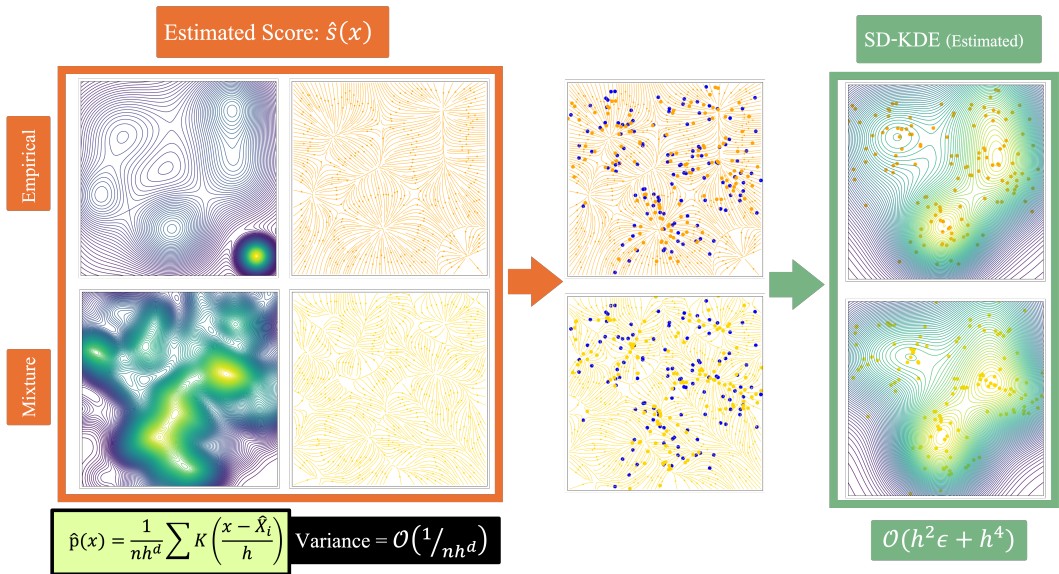

Figure 5: Second schematic diagram on SD-KDE. In the case where a score function is not available. We can use a proxy score function from a proxy distribution. In the example, this is the mixture of the original distribution with some Gaussian distribution. We can also estimate the score from data points. If the estimated score and the actual score are close enough as in the corollary 1, then one can attain a better result with SD-KDE compared to vanilla KDE.

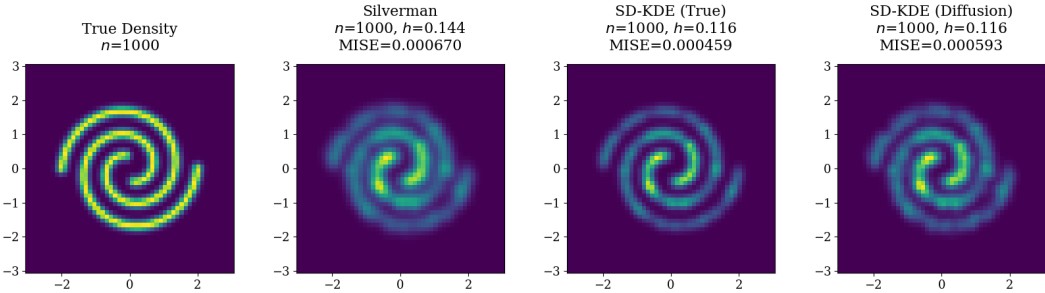

Figure 6: Comparison of the true 2D spiral density vs. Silverman and SD-KDE. For SD-KDE, we evaluate with both the true and learned (diffusion model) score. SD-KDE outperforms Silverman with the oracle score function, and achieves comparable performance even using a noisy score.

score-based correction step, with scale $0.015$ decaying at a rate of $0.15$ at each iteration) to generate surrogate points that remove the leading-order bias. The resulting debiased KDE serves as the baseline for the next iteration, where we recompute the score and apply SD-KDE again; each successive iteration thus leverages a more accurate score estimate to correct residual higher-order biases. Figure 7 shows the method when one iteration is taken. Intuitively, since the first SD-KDE step cancels the dominant bias term, subsequent iterations can target smaller remaining discrepancies, progressively aligning the estimated density more closely with the true distribution. As shown in Figure 7, repeated application of SD-KDE yields a closer alignment between the estimated and true probability densities and a corresponding reduction in KL divergence and mean integrated squared error (MISE) with each iteration. Notably, while a single SD-KDE iteration often captures the majority of the improvement in simpler mixture scenarios (additional iterations confer negligible benefit), more complex multi-modal cases or smaller sample regimes benefit from multiple iterations, albeit with diminishing returns. These results illustrate how SD-KDE could be used to directly improve upon KDE without training a separate score oracle (which can often be difficult to train). Similar to the previous sections, we include a similar analysis for Laplace mixture models in Appendix A.

Table 1: Parameters for the three univariate Gaussian mixtures used in our experiments. Each mixture follows the generic form $p(x) = \pi \mathcal{N}(x \mid \mu_1, \sigma_1^2) + (1 - \pi) \mathcal{N}(x \mid \mu_2, \sigma_2^2)$.

| Mixture | $\pi$ | $\mu_1$ | $\sigma_1$ | $\mu_2$ | $\sigma_2$ |
|---------|-------|---------|------------|---------|------------|
| **1** | 0.4 | -2.0 | 0.5 | 2.0 | 1.0 |
| **2** | 0.3 | -2.0 | 0.4 | 4.0 | 1.5 |
| **3** | 0.5 | 0.0 | 0.4 | 1.5 | 1.5 |

### 3.4 MNIST Dataset

In this study, we follow a similar experimental setup to Liu et al. (2020) and explore the relationship between generated image quality and estimated density using the MNIST dataset—a widely recognized benchmark comprising 70,000 grayscale images ($28 \times 28$ pixels) of handwritten digits (LeCun & Cortes, 2010). We trained a DDPM on this dataset and, by selecting the lowest diffusion timestep ($t = 1$), obtained an estimate of the score function for individual images. Using this score, we apply SD-KDE in latent space to assess the realism of generated images. We ranked generated images from highest to lowest estimated probability density, visualized in Figure 16. The images with higher density appear more realistic and are correlated with higher quality.

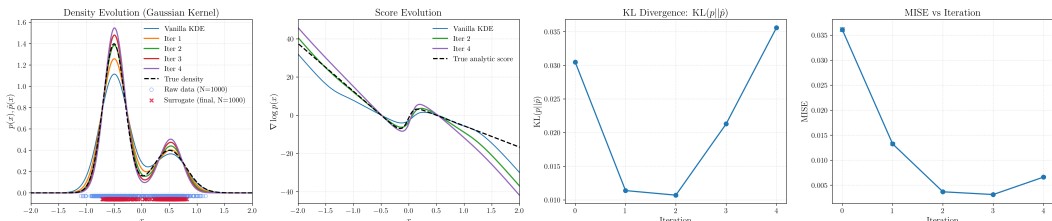

Figure 7: Left to right: (a) Density estimates obtained by vanilla KDE (blue) and by SD-KDE after one to four score-debiased iterations (warm colors). The surrogate samples produced by the final iteration (red, $n = 1000$) visibly sharpen the bimodal structure relative to the raw data (blue, $n = 1000$). (b) The corresponding score functions converge toward the analytic score (black dashed), illustrating progressive removal of higher-order bias. (c) Kullback–Leibler divergence falls by more than a factor of three after the first correction and attains its minimum at the second iteration before mild over-correction appears. (d) Monte-Carlo MISE (mean integrated square error over 200 replicates) mirrors the KL trend, confirming that a small number of SD-KDE steps yields the best bias–variance trade-off for this 1D Gaussian mixture.

## 4 Proof of Theorem 1

*Proof.* First, we decompose the MISE into the bias and the variance terms as

$$\text{MISE} = \int \left( \mathbb{E}\left[\hat{p}(x)\right] - p(x) \right)^2 dx + \int \left( \mathbb{E}\left[\hat{p}(x)^2\right] - \mathbb{E}\left[\hat{p}(x)\right]^2 \right) dx$$

$$= \int \text{Bias}\left[\hat{p}(x)\right]^2 dx + \int \text{Var}\left[\hat{p}(x)\right] dx,$$

where the variance term $\text{Var}\left[\hat{p}(x)\right] = \mathbb{E}\left[\hat{p}(x)^2\right] - \mathbb{E}\left[\hat{p}(x)\right]^2$ and the bias term $\text{Bias}\left[\hat{p}(x)\right] = p(x) - \mathbb{E}\left[\hat{p}(x)\right]$.

The variance term is given by

$$\text{Var}\left[\hat{p}(x)\right] = \frac{1}{n}\text{Var}\left( \frac{1}{h^d} K\left( \frac{x - (X + \delta \hat{s}(X))}{h} \right) \right)$$

since $\hat{p}(x)$ is a sum of $n$ i.i.d. terms. Using Taylor expansion at the kernel $K$ around $\frac{x-X}{h}$ yields

$$K\left( \frac{x - (X + \delta \hat{s}(X))}{h} \right) = K\left( \frac{x - X}{h} \right) - \frac{\delta}{h} \hat{s}(X)^\top \nabla K\left( \frac{x - X}{h} \right) + O\left( \frac{\delta^2}{h^2} \right).$$

The variance is dominated by the leading order term, which gives

$$\text{Var}\left[\hat{p}(x)\right] = \frac{1}{n}\text{Var}\left(\frac{1}{h^d}K\left(\frac{x-X}{h}\right)\right) + \mathcal{O}\left(\frac{\delta^2}{nh^{2+d}}\right) = \mathcal{O}\left(\frac{1}{nh^d} + \frac{\delta^2}{nh^{d+2}}\right)$$

where we used the standard KDE variance result for the leading term.

Now, we analyze the bias term.

$$\text{Bias}[\hat{p}(x)] = \mathbb{E}[\hat{p}(x)] - p(x).$$

We write the expectation of $\hat{p}(x)$ as

$$\mathbb{E}\left[\hat{p}(x)\right] = \frac{1}{h^d}\mathbb{E}\left[K\left(\frac{x-(X+\delta\hat{s}(X))}{h}\right)\right] = \int \frac{1}{h^d}K\left(\frac{x-(y+\delta\hat{s}(y))}{h}\right)p(y)dy.$$

We substitute $u = \frac{x-y}{h}$ to obtain

$$\mathbb{E}\left[\hat{p}(x)\right] = \int K\left(u - \frac{\delta}{h}\hat{s}(x-hu)\right)p(x-hu)du. \tag{1}$$

Taylor expansion will yield that

$$p(x-hu) = p(x) - hu^\top\nabla p(x) + \frac{h^2}{2}u^\top\nabla^2 p(x)u + \mathcal{O}\left(h^3\right),$$

and that

$$K\left(u - \frac{\delta}{h}\hat{s}(x-hu)\right)$$

$$= K\left(u - \frac{\delta}{h}\hat{s}(x) + \delta u^\top\nabla\hat{s}(x) + \mathcal{O}\left(\delta h\right)\right)$$

$$= K(u) - \frac{\delta}{h}\hat{s}(x)^\top\nabla K(u) + \delta u^T\nabla\hat{s}(x)\nabla K(u) + \frac{\delta^2}{2h^2}\hat{s}(x)^\top\nabla^2 K(u)\hat{s}(x) + \mathcal{O}\left(\delta h + \frac{\delta^2}{h} + \frac{\delta^3}{h^3}\right).$$

Substitute these expansions into Equation (1), and expand the product. We consider each term separately.

1. $K(u)p(x)$ integrates to $p(x)$ by the definition of $K$.

2. $-\frac{\delta}{h}\hat{s}(x)^\top\nabla K(u)p(x)$ integrates to $0$ since $K$ is symmetric and decays to $0$ at infinity.

3. $K(u)\left(-hu^\top\nabla p(x)\right)$ integrates to $0$ by the symmetry of $K$.

4. $K(u)\left(\frac{h^2}{2}u^\top\nabla^2 p(x)u\right)$ integrates to $\frac{h^2}{2}\text{Tr}\left(\nabla^2 p(x)\int uu^\top K(u)du\right) = \frac{h^2}{2}\nabla^2 p(x)$.

5. $-\frac{\delta}{h}\hat{s}(x)^\top\nabla K(u)\cdot(-hu^\top\nabla p(x))$. Integrate by parts on $\nabla K(u)$ to obtain

$$\delta\hat{s}(x)^\top\int u^\top\nabla p(x)\nabla K(u)du = \delta\hat{s}(x)^\top\left(-\int K(u)\nabla p(x)du\right) = -\delta\hat{s}(x)^\top\nabla p(x).$$

Using $\hat{s}(x) = \nabla\log p(x)$, we have

$$-\delta\hat{s}(x)^\top\nabla p(x) = -\delta\nabla\log p(x)^\top\nabla p(x) = -\delta\frac{\|\nabla p(x)\|^2}{p(x)}.$$

Using a standard multivariable calculus identity, we have

$$-\delta\frac{\|\nabla p(x)\|^2}{p(x)} = -\delta(\nabla^2 p(x) - p(x)\nabla^2(\log p(x))).$$

6. $p(x)\delta u^T\nabla\hat{s}(x)\nabla K(u)$. Again, after integration by parts, we obtain

$$\delta\int u^T\nabla\hat{s}(x)\nabla K(u)p(x)du = \delta\int p(x)K(u)\text{tr}\left(\nabla\hat{s}(x)\right)du = \delta p(x)\text{tr}\left(\nabla\hat{s}(x)\right) = \delta p(x)\nabla^2(\log p(x)).$$

7. $p(x)\frac{\delta^2}{2h^2}\hat{s}(x)^\top\nabla^2 K(u)\hat{s}(x)$ integrates to 0.

Using smoothness, we then have that

$$\mathbb{E}\left[\hat{p}(x)\right] - p(x) = \frac{h^2}{2}\nabla^2 p(x) - \delta\nabla^2 p(x) + \mathcal{O}\left(h^3 + \delta h + \frac{\delta^2}{h} + \frac{\delta^3}{h^3}\right)$$

Now, by choosing $\delta = \frac{h^2}{2}$, we make the leading term zero, and the bias $\mathbb{E}\left[\hat{p}(x)\right] - p(x) = \mathcal{O}\left(h^3 + \delta h + \frac{\delta^2}{h} + \frac{\delta^3}{h^3}\right) = \mathcal{O}\left(h^3\right)$. Using the standard KDE argument (symmetry of $K$ and decay to 0 at infinity), we can show that $h^3$ terms in the bias also vanish. Thus, the bias is $\mathcal{O}\left(h^4\right)$.

Moreover, note that $\delta = \frac{h^2}{2}$, so the variance term $\text{Var}\left[\hat{p}(x)\right] = \mathcal{O}\left(\frac{1}{nh^d}\right)$.

For optimal error scaling, we balance the bias and the leading variance terms. The error due to bias is $\mathcal{O}\left(h^8\right)$, and the leading error due to variance is $\mathcal{O}\left(\frac{1}{nh^d}\right)$.

Balancing these terms, we obtain $h_{\text{opt}} = \mathcal{O}\left(n^{-1/(d+8)}\right)$.

Finally, the MISE is $\mathcal{O}\left(h^8\right) = \mathcal{O}\left(n^{-8/(d+8)}\right)$. $\qquad\square$

## 5 Additional Discussion

**Connections to Langevin dynamics.** We note that the algorithm is an analog to the continuous time Langevin dynamics, which uses the score function $s$ and yields that the stochastic differential equation

$$dX_t = \frac{1}{2}s\left(X_t\right)dt + dB_t \tag{2}$$

will have the stationary distribution according to the probability distribution function $p$, which corresponds to the score function $s$ (Song & Ermon, 2019; Song et al., 2020). Our work can be viewed as a one-step Euler–Maruyama discretization of the Langevin dynamics to estimate the location-shifted kernel from the sample points. This ensures both tractability as well as the benefit of bias-reduction as seen in the main theorem (Theorem 1). To our knowledge, this is the first approach that employs Langevin dynamics to inform a position-based debiased kernel density estimator.

**Bridging Score-Based and Sample-Based Density Estimation.** While the paper (Song et al., 2020) suggests a formulation of the flow ODE as an evolution of the density function from an approximate posterior. This approach is prior-free and the flow maps a scaled Gaussian distribution to the data distribution. However, this process does not utilize the availability of samples and relies solely on the score estimate. Since this scheme requires spatial and temporal discretization for density estimation, it is computationally less feasible due to the curse of dimensionality.

Many works in non-parametric methods (ie. KDE, histogram) (Silverman, 1986; Rosenblatt, 1956; Parzen, 1962; Scott, 1979; Lugosi & Nobel, 1996) and neural-based density estimation (Liu et al., 2021; Magdon-Ismail & Atiya, 1998; Rezende & Mohamed, 2015; Dinh et al., 2016; Berg et al., 2018) use the sample points for the density estimation, but do not incorporate score function in the density estimation framework.

A promising future direction is to consider a multi-step discretization of the Langevin dynamics to obtain asymptotically superior debiasing. Using higher order discretization schemes is also an interesting avenue that we are currently exploring. The multi-step approach introduces more challenges, including non-Gaussianity of the final kernel, since it will be a convolution of multiple Gaussian kernels with different score-dependent shifts.

## 6 Conclusion

In this work, we demonstrate that incorporating score information can asymptotically improve density estimation accuracy. We propose a method for score-debiased kernel density estimation that achieves $\mathcal{O}\left(n^{-8/(d+8)}\right)$ convergence rate in mean integrated squared error, improving upon the classical $\mathcal{O}\left(n^{-4/(d+4)}\right)$ rate of standard KDE. Our experiments on a variety of synthetic datasets validate these theoretical predictions and show that the method remains effective even when using noisy score estimates, suggesting practical applicability beyond settings where the true score is known.

**Limitations.** A key limitation of our proposed method is that the theoretical performance guarantees for SD-KDE require access to an exact score oracle, which is typically unavailable in practical scenarios. Although our empirical results demonstrate that accurate score estimates obtained from state-of-the-art methods (such as score matching or diffusion models) still provide significant performance improvements, these estimation methods themselves can be computationally expensive, particularly in high-dimensional settings or with large datasets. Future work might explore more efficient score estimation techniques or approximate methods that retain the benefits of SD-KDE while reducing the associated computational overhead.

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

# A Synthetic 1D experiments

Figure 8 and Figure 10 shows the fitted densities for different noise levels of the SD-KDE method, as well as the Silverman baseline, for $n = 200$ samples for three different Gaussian (and Laplace respectively) mixture models, with parameters outlined in Table 1. In Figure 11, we examine the consistency of the performance gains for the SD-KDE method over the Silverman baseline for a mixture of Laplace densities. The Laplace mixtures use the same location and scale parameters as the Gaussian Mixture, given in Table 1.

Next, we show the scaling in $n$ for a density estimation task for Laplace Mixtures. Figure 9 shows the results. In Figure 12, and 13, we show a visualization of the score function and the densities for both the Gaussian and Laplace Mixtures.

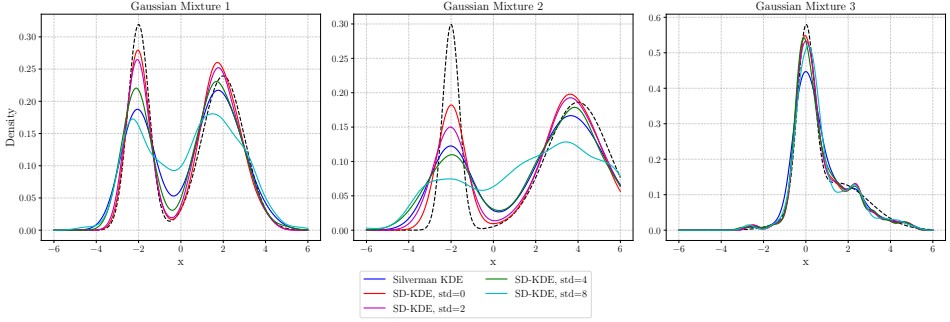

Figure 8: Drawing $n = 200$ samples from each of the three Gaussian mixtures in equation 3.1 The dashed black line is the *true* PDF, while the colored lines represent the estimated PDFs.

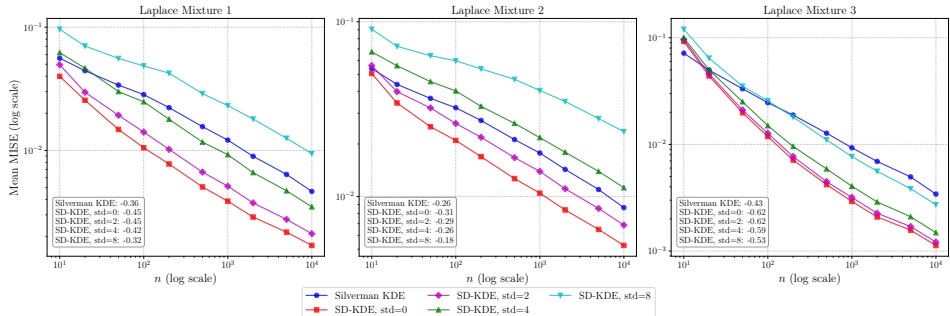

Figure 9: MISE error as a function of $n$ for each of the three gaussian mixtures. For each point, we compute the MISE with 50 random seeds per mixture. Each subplot plots the mean integrated squared error as a function of $n$. The legend compares Silverman KDE to SD-KDE at multiple noise settings. The slopes inside each subplot are fitted regression lines in log–log scale indicating how quickly each method's error decays as $n$ increases.

# B Synthetic 2D mixture of Gaussians

In Figure 15, on a mixture of Gaussians ground-truth density, we compare the Silverman method with SD-KDE.

# C MNIST Dataset Image

The following figure depicts the ordering of generated images based on estimated probability density values.

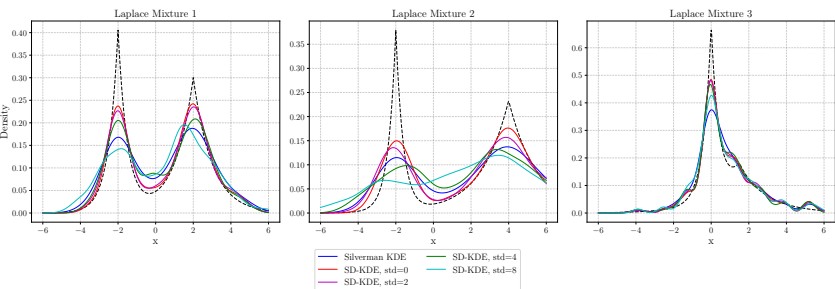

Figure 10: Drawing $n = 200$ samples from each of the three Laplace mixtures in equation 3.1 The dashed black line is the true probability density function, while the colored lines represent the estimated probability density functions.

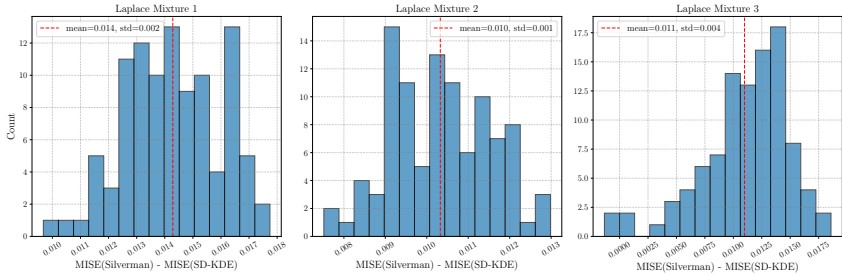

Figure 11: Histogram of MISE difference of the SD-KDE method and the Silverman method, for $n = 100$ samples and 50 random seeds per mixture. A positive value in the plot indicates that the SD-KDE method performed better for that seed. We observe that SD-KDE consistently performs better than the Silverman method over multiple random seeds.

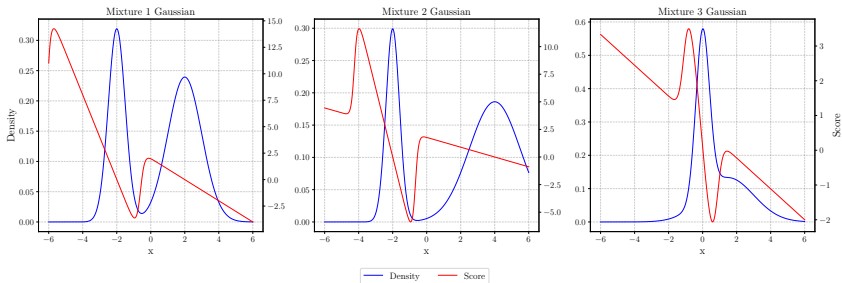

Figure 12: In each subplot, we plot the Gaussian mixture's density (blue, left axis) and the log-density derivative (score) in red (right axis).

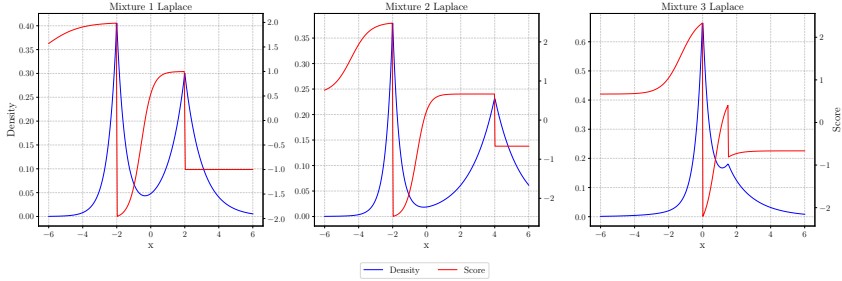

Figure 13: In each subplot, we plot the Laplace mixture's density (blue, left axis) and the log-density derivative (score) in red (right axis).

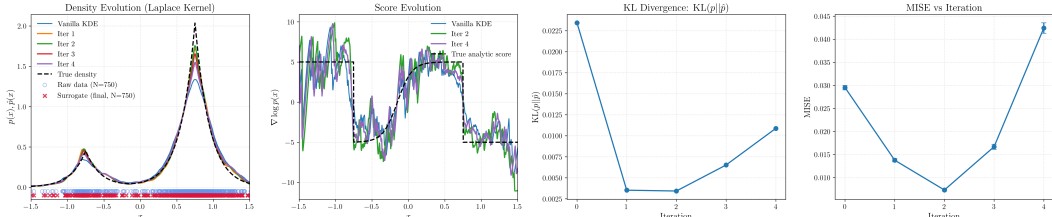

Figure 14: Left to right: (a) Density estimates obtained by vanilla KDE (blue) and by SD-KDE after one to four score-debiased iterations (warm colours). The surrogate samples produced by the final iteration (red, $n = 1000$) visibly sharpen the bimodal structure relative to the raw data (blue, $n = 1000$). (b) The corresponding score functions converge toward the analytic score (black dashed), illustrating progressive removal of higher-order bias. (c) Kullback–Leibler divergence falls by more than a factor of three after the first correction and attains its minimum at the second iteration before mild over-correction appears. (d) Monte-Carlo MISE (mean integrated square error over 200 replicates) mirrors the KL trend, confirming that a small number of SD-KDE steps yields the best bias–variance trade-off for this 1D Laplacian mixture.

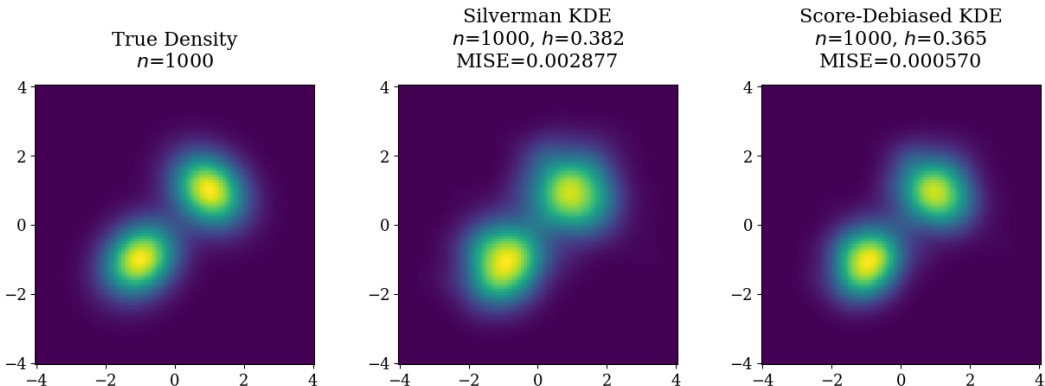

Figure 15: Comparison of a true 2D mixture of Gaussians density vs. the Silverman method and our SD-KDE method using the true score. Given the oracle score function, SD-KDE outperforms Silverman in MISE by nearly an order of magnitude.


Figure 16: Generated MNIST images of digits 2, 3, and 7 are displayed in descending order of estimated probability density as determined by score-based KDE. The ordering illustrates that images with higher probability density estimates exhibit more realistic features.

2. **Limitations**

   Question: Does the paper discuss the limitations of the work performed by the authors?

   Answer: [Yes]

   Justification: We have a limitation section which addresses the strong assumptions made, and the difficulties that come with these assumptions. See Section 6.

   Guidelines:

   - The answer NA means that the paper has no limitation while the answer No means that the paper has limitations, but those are not discussed in the paper.
   - The authors are encouraged to create a separate "Limitations" section in their paper.
   - The paper should point out any strong assumptions and how robust the results are to violations of these assumptions (e.g., independence assumptions, noiseless settings, model well-specification, asymptotic approximations only holding locally). The authors should reflect on how these assumptions might be violated in practice and what the implications would be.
   - The authors should reflect on the scope of the claims made, e.g., if the approach was only tested on a few datasets or with a few runs. In general, empirical results often depend on implicit assumptions, which should be articulated.
   - The authors should reflect on the factors that influence the performance of the approach. For example, a facial recognition algorithm may perform poorly when image resolution is low or images are taken in low lighting. Or a speech-to-text system might not be used reliably to provide closed captions for online lectures because it fails to handle technical jargon.
   - The authors should discuss the computational efficiency of the proposed algorithms and how they scale with dataset size.
   - If applicable, the authors should discuss possible limitations of their approach to address problems of privacy and fairness.
   - While the authors might fear that complete honesty about limitations might be used by reviewers as grounds for rejection, a worse outcome might be that reviewers discover limitations that aren't acknowledged in the paper. The authors should use their best judgment and recognize that individual actions in favor of transparency play an important role in developing norms that preserve the integrity of the community. Reviewers will be specifically instructed to not penalize honesty concerning limitations.

3. **Theory assumptions and proofs**

   Question: For each theoretical result, does the paper provide the full set of assumptions and a complete (and correct) proof?

   Answer: [Yes]

Justification: Yes, we have a single theorem and that has a correct proof.

Guidelines:

- The answer NA means that the paper does not include theoretical results.
- All the theorems, formulas, and proofs in the paper should be numbered and cross-referenced.
- All assumptions should be clearly stated or referenced in the statement of any theorems.
- The proofs can either appear in the main paper or the supplemental material, but if they appear in the supplemental material, the authors are encouraged to provide a short proof sketch to provide intuition.
- Inversely, any informal proof provided in the core of the paper should be complemented by formal proofs provided in appendix or supplemental material.
- Theorems and Lemmas that the proof relies upon should be properly referenced.

4. **Experimental result reproducibility**

Question: Does the paper fully disclose all the information needed to reproduce the main experimental results of the paper to the extent that it affects the main claims and/or conclusions of the paper (regardless of whether the code and data are provided or not)?

Answer: [Yes]

Justification: Yes, the details of the experiments are all laid out in the document. Any datasets which were used are either publicly available or easy to generate by following the the paper.

Guidelines:

- The answer NA means that the paper does not include experiments.
- If the paper includes experiments, a No answer to this question will not be perceived well by the reviewers: Making the paper reproducible is important, regardless of whether the code and data are provided or not.
- If the contribution is a dataset and/or model, the authors should describe the steps taken to make their results reproducible or verifiable.
- Depending on the contribution, reproducibility can be accomplished in various ways. For example, if the contribution is a novel architecture, describing the architecture fully might suffice, or if the contribution is a specific model and empirical evaluation, it may be necessary to either make it possible for others to replicate the model with the same dataset, or provide access to the model. In general. releasing code and data is often one good way to accomplish this, but reproducibility can also be provided via detailed instructions for how to replicate the results, access to a hosted model (e.g., in the case of a large language model), releasing of a model checkpoint, or other means that are appropriate to the research performed.
- While NeurIPS does not require releasing code, the conference does require all submissions to provide some reasonable avenue for reproducibility, which may depend on the nature of the contribution. For example
  (a) If the contribution is primarily a new algorithm, the paper should make it clear how to reproduce that algorithm.
  (b) If the contribution is primarily a new model architecture, the paper should describe the architecture clearly and fully.
  (c) If the contribution is a new model (e.g., a large language model), then there should either be a way to access this model for reproducing the results or a way to reproduce the model (e.g., with an open-source dataset or instructions for how to construct the dataset).
  (d) We recognize that reproducibility may be tricky in some cases, in which case authors are welcome to describe the particular way they provide for reproducibility. In the case of closed-source models, it may be that access to the model is limited in some way (e.g., to registered users), but it should be possible for other researchers to have some path to reproducing or verifying the results.

5. **Open access to data and code**

Question: Does the paper provide open access to the data and code, with sufficient instructions to faithfully reproduce the main experimental results, as described in supplemental material?

Answer: [Yes]

Justification: Yes, we provide code for all experiments in the supplementary material.

Guidelines:

- The answer NA means that paper does not include experiments requiring code.
- Please see the NeurIPS code and data submission guidelines (`https://nips.cc/public/guides/CodeSubmissionPolicy`) for more details.
- While we encourage the release of code and data, we understand that this might not be possible, so "No" is an acceptable answer. Papers cannot be rejected simply for not including code, unless this is central to the contribution (e.g., for a new open-source benchmark).
- The instructions should contain the exact command and environment needed to run to reproduce the results. See the NeurIPS code and data submission guidelines (`https://nips.cc/public/guides/CodeSubmissionPolicy`) for more details.
- The authors should provide instructions on data access and preparation, including how to access the raw data, preprocessed data, intermediate data, and generated data, etc.
- The authors should provide scripts to reproduce all experimental results for the new proposed method and baselines. If only a subset of experiments are reproducible, they should state which ones are omitted from the script and why.
- At submission time, to preserve anonymity, the authors should release anonymized versions (if applicable).
- Providing as much information as possible in supplemental material (appended to the paper) is recommended, but including URLs to data and code is permitted.

6. **Experimental setting/details**

Question: Does the paper specify all the training and test details (e.g., data splits, hyperparameters, how they were chosen, type of optimizer, etc.) necessary to understand the results?

Answer: [Yes]

Justification: All the parameters, such as the noise level or the bandwidth were specified. All the training details, such as the dataset and which parameters for mixture models were used, were specified.

Guidelines:

- The answer NA means that the paper does not include experiments.
- The experimental setting should be presented in the core of the paper to a level of detail that is necessary to appreciate the results and make sense of them.
- The full details can be provided either with the code, in appendix, or as supplemental material.

7. **Experiment statistical significance**

Question: Does the paper report error bars suitably and correctly defined or other appropriate information about the statistical significance of the experiments?

Answer: [Yes]

Justification: We include error bars in key Figures such as Figure 7.

Guidelines:

- The answer NA means that the paper does not include experiments.
- The authors should answer "Yes" if the results are accompanied by error bars, confidence intervals, or statistical significance tests, at least for the experiments that support the main claims of the paper.
- The factors of variability that the error bars are capturing should be clearly stated (for example, train/test split, initialization, random drawing of some parameter, or overall run with given experimental conditions).

- The method for calculating the error bars should be explained (closed form formula, call to a library function, bootstrap, etc.)
- The assumptions made should be given (e.g., Normally distributed errors).
- It should be clear whether the error bar is the standard deviation or the standard error of the mean.
- It is OK to report 1-sigma error bars, but one should state it. The authors should preferably report a 2-sigma error bar than state that they have a 96% CI, if the hypothesis of Normality of errors is not verified.
- For asymmetric distributions, the authors should be careful not to show in tables or figures symmetric error bars that would yield results that are out of range (e.g. negative error rates).
- If error bars are reported in tables or plots, The authors should explain in the text how they were calculated and reference the corresponding figures or tables in the text.

8. **Experiments compute resources**

   Question: For each experiment, does the paper provide sufficient information on the computer resources (type of compute workers, memory, time of execution) needed to reproduce the experiments?

   Answer: [Yes]

   Justification: Yes, we explicitly provided the computing hardware and runtime in 3.1.

   Guidelines:

   - The answer NA means that the paper does not include experiments.
   - The paper should indicate the type of compute workers CPU or GPU, internal cluster, or cloud provider, including relevant memory and storage.
   - The paper should provide the amount of compute required for each of the individual experimental runs as well as estimate the total compute.
   - The paper should disclose whether the full research project required more compute than the experiments reported in the paper (e.g., preliminary or failed experiments that didn't make it into the paper).

9. **Code of ethics**

   Question: Does the research conducted in the paper conform, in every respect, with the NeurIPS Code of Ethics https://neurips.cc/public/EthicsGuidelines?

   Answer: [Yes]

   Justification: Yes, the paper conforms with the NeurIPS Code of Ethics.

   Guidelines:

   - The answer NA means that the authors have not reviewed the NeurIPS Code of Ethics.
   - If the authors answer No, they should explain the special circumstances that require a deviation from the Code of Ethics.
   - The authors should make sure to preserve anonymity (e.g., if there is a special consideration due to laws or regulations in their jurisdiction).

10. **Broader impacts**

    Question: Does the paper discuss both potential positive societal impacts and negative societal impacts of the work performed?

    Answer: [NA]

    Justification: The paper is on a novel method for density estimation. It is very far removed from any societal concerns.

    Guidelines:

    - The answer NA means that there is no societal impact of the work performed.
    - If the authors answer NA or No, they should explain why their work has no societal impact or why the paper does not address societal impact.

- Examples of negative societal impacts include potential malicious or unintended uses (e.g., disinformation, generating fake profiles, surveillance), fairness considerations (e.g., deployment of technologies that could make decisions that unfairly impact specific groups), privacy considerations, and security considerations.
- The conference expects that many papers will be foundational research and not tied to particular applications, let alone deployments. However, if there is a direct path to any negative applications, the authors should point it out. For example, it is legitimate to point out that an improvement in the quality of generative models could be used to generate deepfakes for disinformation. On the other hand, it is not needed to point out that a generic algorithm for optimizing neural networks could enable people to train models that generate Deepfakes faster.
- The authors should consider possible harms that could arise when the technology is being used as intended and functioning correctly, harms that could arise when the technology is being used as intended but gives incorrect results, and harms following from (intentional or unintentional) misuse of the technology.
- If there are negative societal impacts, the authors could also discuss possible mitigation strategies (e.g., gated release of models, providing defenses in addition to attacks, mechanisms for monitoring misuse, mechanisms to monitor how a system learns from feedback over time, improving the efficiency and accessibility of ML).

11. **Safeguards**

Question: Does the paper describe safeguards that have been put in place for responsible release of data or models that have a high risk for misuse (e.g., pretrained language models, image generators, or scraped datasets)?

Answer: [NA]

Justification: The paper poses no such risks.

Guidelines:

- The answer NA means that the paper poses no such risks.
- Released models that have a high risk for misuse or dual-use should be released with necessary safeguards to allow for controlled use of the model, for example by requiring that users adhere to usage guidelines or restrictions to access the model or implementing safety filters.
- Datasets that have been scraped from the Internet could pose safety risks. The authors should describe how they avoided releasing unsafe images.
- We recognize that providing effective safeguards is challenging, and many papers do not require this, but we encourage authors to take this into account and make a best faith effort.

12. **Licenses for existing assets**

Question: Are the creators or original owners of assets (e.g., code, data, models), used in the paper, properly credited and are the license and terms of use explicitly mentioned and properly respected?

Answer: [Yes]

Justification: All assets are original or explicitly mentioned/properly cited.

Guidelines:

- The answer NA means that the paper does not use existing assets.
- The authors should cite the original paper that produced the code package or dataset.
- The authors should state which version of the asset is used and, if possible, include a URL.
- The name of the license (e.g., CC-BY 4.0) should be included for each asset.
- For scraped data from a particular source (e.g., website), the copyright and terms of service of that source should be provided.
- If assets are released, the license, copyright information, and terms of use in the package should be provided. For popular datasets, `paperswithcode.com/datasets` has curated licenses for some datasets. Their licensing guide can help determine the license of a dataset.

- For existing datasets that are re-packaged, both the original license and the license of the derived asset (if it has changed) should be provided.
- If this information is not available online, the authors are encouraged to reach out to the asset's creators.

13. **New assets**

Question: Are new assets introduced in the paper well documented and is the documentation provided alongside the assets?

Answer: [Yes]

Justification: Any of the training data which is not public is synthetic, and we detailed clearly how to reproduce it. The code is documented in the aforementioned released version.

Guidelines:

- The answer NA means that the paper does not release new assets.
- Researchers should communicate the details of the dataset/code/model as part of their submissions via structured templates. This includes details about training, license, limitations, etc.
- The paper should discuss whether and how consent was obtained from people whose asset is used.
- At submission time, remember to anonymize your assets (if applicable). You can either create an anonymized URL or include an anonymized zip file.

14. **Crowdsourcing and research with human subjects**

Question: For crowdsourcing experiments and research with human subjects, does the paper include the full text of instructions given to participants and screenshots, if applicable, as well as details about compensation (if any)?

Answer: [NA]

Justification: This paper does not involve crowdsourcing nor research with human subjects.

Guidelines:

- The answer NA means that the paper does not involve crowdsourcing nor research with human subjects.
- Including this information in the supplemental material is fine, but if the main contribution of the paper involves human subjects, then as much detail as possible should be included in the main paper.
- According to the NeurIPS Code of Ethics, workers involved in data collection, curation, or other labor should be paid at least the minimum wage in the country of the data collector.

15. **Institutional review board (IRB) approvals or equivalent for research with human subjects**

Question: Does the paper describe potential risks incurred by study participants, whether such risks were disclosed to the subjects, and whether Institutional Review Board (IRB) approvals (or an equivalent approval/review based on the requirements of your country or institution) were obtained?

Answer: [NA]

Justification: This paper does not involve crowdsourcing nor research with human subjects.

Guidelines:

- The answer NA means that the paper does not involve crowdsourcing nor research with human subjects.
- Depending on the country in which research is conducted, IRB approval (or equivalent) may be required for any human subjects research. If you obtained IRB approval, you should clearly state this in the paper.
- We recognize that the procedures for this may vary significantly between institutions and locations, and we expect authors to adhere to the NeurIPS Code of Ethics and the guidelines for their institution.

- For initial submissions, do not include any information that would break anonymity (if applicable), such as the institution conducting the review.

16. **Declaration of LLM usage**

    Question: Does the paper describe the usage of LLMs if it is an important, original, or non-standard component of the core methods in this research? Note that if the LLM is used only for writing, editing, or formatting purposes and does not impact the core methodology, scientific rigorousness, or originality of the research, declaration is not required.

    Answer: [Yes]

    Justification: An LLM was used in the ideation phase in the proof of the theorem.

    Guidelines:

    - The answer NA means that the core method development in this research does not involve LLMs as any important, original, or non-standard components.
    - Please refer to our LLM policy (`https://neurips.cc/Conferences/2025/LLM`) for what should or should not be described.

