# OpenReview forum: "SD-KDE: Score-Debiased Kernel Density Estimation"
_NeurIPS.cc/2025/Conference — NeurIPS 2025 poster_

### Official Review · Reviewer_4YAs · 2025-06-11

**Clarity:** 3
**Significance:** 2
**Originality:** 2
**Rating:** 4
**Confidence:** 4

**Summary:**

Kernel density estimation is a classical density estimation technique which yields properly normalised nonparametric density estimates. Score based methods match only the score function (gradient of log density), which does not require the normalising constant, and therefore does not yield a density estimate in a classical sense. The authors propose to use information in (an estimate of) the score function to assist in KDE, yielding properly normalised score debiased KDE (SD-KDE). This is achieved through updating the original data samples in one step of the gradient of the score function. A theoretical result shows improved estimation error compared with naive KDE, as well as optimal bandwidth and step size selection, in the case where the exact score function is known. Experiments are demonstrated on a range of low dimensional settings, showing good benefits above classical KDE.

**Questions:**

I am currently borderline between reject and accept. I err on the positive side (believing in advance that the authors will give a reasonable rebuttal) and give accept for this review. I am unlikely to raise my score due to the main weakness of corollary 1, unless the authors can somehow improve the result or apply it to a concrete setting. Actionable summary of my weaknesses:

- Please discuss whether something more practical can be salvaged from Corollary 1. For example, is it possible to upper bound the integrated RHS? Are there any takeaways for practical score estimation?
- Please answer my question about whether independent training set is used for score matching versus the KDE step. Include any modifications you might make to an updated manuscript.
- Please address the minor comments.

**Ethical Concerns:**

["NO or VERY MINOR ethics concerns only"]

**Final Justification:**

I previously noted that I would maintain my score, if the authors gave a good rebuttal. I'm satisfied with their response to my questions and to the points raised by other reviewers.

It seems that most other reviewers find this an interesting and mostly sound paper (with minor improvements pointed out in the reviews --- different ones each caught by different reviewers), sufficient for NeurIPS.

I am puzzled by Reviewer cUSv's review. Perhaps they have valid points but perhaps not. Given they have not engaged in the discussion so far, I am inclined to make a hard (binary) decision to disregard their review in my final justification.

UPDATE
I agree that there is a gap between knowing the score function and estimating the score function, and the paper does not address this gap. A much stronger paper would include a result that accounts for inexact score. (Bear in mind, knowing the score function does not mean knowing the normalising constant. For example, we can estimate the posterior of a Bayesian model with the exact score function using this method). However, all things considered, I am still inclined to borderline accept the paper, because it offers quite a new perspective.

At the end of the day, I will not be upset if the AC decides to reject the paper.

**Limitations:**

Yes.

Limitations are pointed out in the paper, and agree with the major weakness in my review above. I am not concerned about negative societal impact for this broad application-agnostic paper.

**Paper Formatting Concerns:**

None.

The formatting in section 4 is a little bit unusual, but not enough to exclude publication.

**Quality:**

3

**Strengths And Weaknesses:**

**Strengths:**
- Algorithm 1 is very elegant and clean. It is easy to read and implement. The intuitive description provided by the authors about using the score to debias the samples is helpful.
- I liked that the authors showed two demonstrations of approximate score functions. One involves an initial KDE and then using the score, and the other involves the more standard neural estimator via Hyvarinen score or Fisher divergence. (Unforunately, the neural estimator seems to perform not so well, but this is a separate point).

**Weaknesses:**
- The presentation of theorem 1 could be improved, in my opinion.
    - First, there is no need for the first equation, as it is just substituting step 1 into step 2 of algorithm 1, which is already referenced in the theorem text.
    - Second, I believe "asymptotically optimal" is more precisely "the smallest AMISE". Please correct me if I'm wrong.
    - Third, "Let $\hat{s}$ be the exact score function of $p$." could more precisely be stated as "Let $\hat{s}$ be the exact score function of $p$, i.e. $\hat{s} = s$".
- There is a gap between Corollary 1 and a notion of "error" obtained by standard scoring algorithms. Unfortunately, I can't really see how corollary 1 is useful in its current form. It would be nicer if we could relate the AMISE to some error in the score estimate (e.g. the error in the Hyvarinen matching). This I see as the biggest weakness of this paper. This is acknowledged in the limitations section, by the authors. Would it be possible to upper bound the first term in the RHS of corollary 1?
- Figure 16 is qualitative and it is difficult to assess its meaning. I am not convinced that the images on the right are less representative than the images on the left (I don't have any suggestions for improvement, unfortunately).
- The proof of theorem 1 reveals that some conditions are missing from the statement of theorem 1. Namely, the kernel has to have a convergent Taylor series (satisfying Taylor's theorem)

**Questions:**
- Does one (or should one) use the same datapoints for both the score estimate and the SD-KDE, or should you use different points? Would it make sense to first estimate the score using an independent pretraining set? Why or why not, and did you try both ways?

Minor:
- There are three counts of "the theorem 1". Recommend changing to "Theorem 1", to retain consistency throughout.
- Line 79 uses the nonstandard notaiton $\mathcal{N}(0, \sigma)$, which maybe should be $\mathcal{N}(0, \sigma^2)$.
- I believe "std" in the legends in Figure 2 should be $\sigma$, however this is not mentioned in the text.

---

> ### Author Rebuttal · Authors · 2025-07-31
>
> We would like to thank reviewer 4YAs for their thoughtful review, which has made our paper better.
>
> W1 (presentation of theorem 1): Thank you for these suggestions, we will update as you suggested to make the presentation more clear and concise.
>
> W2 (corollary 1): The main intention of corollary 1 is to give some intuition on how the bias decompose when we only have an estimate for the score. In the two terms one is proportional to the gradient of p, which will be small near modes, and the other is proportional to p, and will be small in the tails of the distribution. Taken together, we can get practical intuition that the bias may concentrate more on moderate density regions rather than extreme trail or modes. We thank the reviewer for pointing out that this is not central to the paper, and we will hence move it to the appendix for the camera ready.
>
> W3 (MNIST figure): In our revision, we will quantitative evaluations to better support the MNIST results. Specifically, we use Classifier confidence: The average top-1 confidence, \frac{1}{N} \sum_{i=1}^{N} \max_{k} \, P(y = k \mid \text{image}_i),  from a pretrained MNIST classifier on generated samples, which serves as a proxy for sample realism. This is more clearly showing the advantage of our method.
>
> W4 (convergent taylor series condition theorem 1): Thank you for pointing this out, we have updated the script with this condition.
>
> Q1 (same vs different points SD-KDE): In practical settings, often only one dataset of points is provided, hence in our experiments with using the KDE to estimate the Score for SD-KDE, we estimated the score from the same set of datapoints as was used for SD-KDE.
> In the updated manuscript we will include a detailed analysis of the tradeoff between using independent vs overlapping samples for the estimates.
>
> Minor comments:
> Thank you for pointing out the formatting errors, indeed, N(0,σ) should be N(0,σ^2), we will use Theorem throughout, and will update the legend in figure 2 to use σ instead of std for the camera ready.

---

### Official Review · Reviewer_cUSv · 2025-07-02

**Clarity:** 2
**Significance:** 1
**Originality:** 1
**Rating:** 2
**Confidence:** 5

**Summary:**

This paper debiased KDE using an estimated score function.

**Questions:**

I don't quite see a clear path for me to raise my score.

**Ethical Concerns:**

["NO or VERY MINOR ethics concerns only"]

**Final Justification:**

My score will remain unchanged. I think a revision would make the paper much stronger.

**Limitations:**

yes.

**Paper Formatting Concerns:**

N/A.

**Quality:**

2

**Strengths And Weaknesses:**

The primary weakness is the lack of novelty and motivation. KDE is non-parametric and not scalable. In its comfort zone, there are many ways to improve the performance of a standard KDE without invoking diffusion models, and of course the silverman baseline. Theorem 1 is also highly idealized.

---

> ### Author Rebuttal · Authors · 2025-07-31
>
> We thank the reviewer cUSv for their review.
>
> We would like to point out that SD-KDE can be applied effectively without invoking diffusion models, in Figure 5 in the paper, SD-KDE is applied with a score estimate from the Silverman KDE.
>
> W1 (lack of novelty/motivation): As described in our paper, kernel density estimation is an important modeling technique in a variety of fields, such as anomaly detection, clustering, data visualization, and dynamical systems. We provide
> 1. A theoretical results for better asymptotic convergence
> 2. A simple algorithm to compute the SD-KDE
> 3. Practical empirical analysis showing the improved performance of the method
>
> Based on this, we hope that the reviewer will reconsider the low scores given to the paper.

---

### Official Review · Reviewer_etfG · 2025-07-02

**Clarity:** 3
**Significance:** 3
**Originality:** 3
**Rating:** 4
**Confidence:** 3

**Summary:**

The paper presents SD-KDE (Score-Debiased Kernel Density Estimation), a method that enhances classical KDE by moving each sample point along the estimated score function before applying KDE with an adjusted bandwidth. This single-step correction cancels the leading-order bias and improves the MISE. The authors provide a complete theoretical analysis and demonstrate consistent performance improvements on synthetic 1D/2D data and MNIST.

**Questions:**

- How does SD-KDE perform in high-dimensional spaces with limited data and noisy scores?
- How does SD-KDE compare with neural density estimators like normalizing flows or autoregressive models in terms of accuracy and efficiency?

**Ethical Concerns:**

["NO or VERY MINOR ethics concerns only"]

**Final Justification:**

**Concerns:** Exact score assumption, few baselines, tuning sensitivity, low-dim focus.
**Response:** Clarified non-learned score use, added MNIST confidence metric and 10D Gaussian mixture, explained $\delta$–$h$ relation and robustness, committed to modern KDE comparisons.
**Outcome:** Maintained borderline accept.

**Limitations:**

yes

**Paper Formatting Concerns:**

no issues

**Quality:**

3

**Strengths And Weaknesses:**

Strengths

- Reduces KDE bias using score-guided data shifts, improving estimation accuracy.
- Achieves provable MISE improvement.
- Experimental results confirm strong gains across settings and noise levels.
- Performs well even when the score function is noisy or estimated from data.
- Iterative application enables further bias reduction without requiring a score oracle.
- Bridges score-based generative modeling and nonparametric density estimation.

Weaknesses

- Theoretical results require access to the exact score function, which is rarely available.
- No empirical comparison with modern KDE variants beyond Silverman’s rule. For instance, recent approaches such as Fast KDE with density matrices and Random Fourier Features (Gallego et al., 2022) and the diffusion-based KDE method diffKDE with optimal bandwidth approximation (Pelz et al., 2023) are not considered.
- Step size $\delta$ and bandwidth $h$ may require careful tuning; robustness not analyzed.
- High-dimensional and structured data domains remain largely untested.
- MNIST results are mostly visual; lacks quantitative metrics for density quality.

---

> ### Author Rebuttal · Authors · 2025-07-31
>
> We would like to thank reviewer for their thoughtful review, which has made our paper better.
>
> W1(Theoretical results require exact score): We believe that it may be possible to extend the analysis in the current work to situations where we don’t know the exact score, this will be the topic of follow up research.
> W2(Visual results on MNIST): In our revision, we have added quantitative evaluations to better support the MNIST results. Specifically, we use Classifier confidence: The average top-1 confidence, \frac{1}{N} \sum_{i=1}^{N} \max_{k} \, P(y = k \mid \text{image}_i),  from a pretrained MNIST classifier on generated samples, which serves as a proxy for sample realism. This is more clearly showing the advantage of our method.
> W3(Tuning of step size and band width): Once the band-width is set, there is a fixed formula for the step size based on the band width. We observed that using a similar constant on the band width term as in the Sliverman method works well without being very sensitive to the exact constant value. We will perform an in-depth analysis in the camera ready version of the paper.
> W4(comparisons with other KDE methods): We want to thank the reviewer for this valuable suggestion, we will compare the SD-KDE method with the more recent methods for the camera ready paper.
>
> Q1(high dimensional setting): Thank you for the suggestion, we have included 10 dimensional gaussian mixture results for the camera ready paper.
> Q2(efficiency comparison with normalizing flow and autoregressive modeling): Compared with these neural approaches, SD-KDE offers a much simpler fitting scheme, where each point is moved in the direction of the score and then applying standard KDE.

---

> > ### Comment · Reviewer_etfG · 2025-08-03
> >
> > Thank you for your clarifications. I will maintain my score.

---

### Official Review · Reviewer_f5yx · 2025-07-02

**Clarity:** 3
**Significance:** 3
**Originality:** 4
**Rating:** 4
**Confidence:** 4

**Summary:**

This paper provides a new kernel based density estimate method that leverages the "score" (the gradient of the log-density, which shows up in diffusion models).  Each data sample is moved in the direction of the score (essentially denoising the samples), then a KDE is built on the updated points.  The paper shows:
   - this significantly increases the rate of estimation from O(n^{-4/(d+4)}) to O(n^{-8/(d+8)})
   - shows improvement in practice in 1 and 2 dimensions (and perhaps on MNIST)
   - if one can only estimate the score, then one can get nearly as much improvement.

**Questions:**

Is it possible to obtain a stronger total bound on the convergence of the density estimate, without assuming the score oracle, but rather estimating the score and using that (as shown to work empirically)?  Are there obvious reasons why this is non-trivial?

**Ethical Concerns:**

["NO or VERY MINOR ethics concerns only"]

**Final Justification:**

The paper makes exciting progress on a classic and important problem.  This is very unexpected.  However it is conditioned on oracle access to the "score" function.  This seems to be easy estimate, but it is not clear in theory, and the emperical experiments (which show improvement) are limited.

It may be an important paper, but some questions remain.

**Limitations:**

yes.

**Quality:**

3

**Strengths And Weaknesses:**

The O(n^{-4/(d+4)}) convergence result (of Silverman 1984) is classic, and so improvement on this is very exciting.  Directly improving it would merit strong acceptance.
The paper falls a bit short of this, since it uses the score as an oracle in the result, and then bounds error in estimated score, and shows how to estimate it in practice (which still shows improvement empirically on Silverman's result, but not as much as with the oracle).
The paper does not provide an improvement in density estimate without knowing this oracle or having some known bound on the accuracy of the score.

Regardless, I find this as very important progress on a classic and very fundamental result, one in which the line of work was thought to have been perhaps closed.
I suspect this will soonafter lead to future work which will close this loop on this approach -- I wish it was in this paper.


So there are many important strengths to the paper, but here are some relatively minor weaknesses:

 - The paper would be improved with better early-on discussion (and mathematical definition) of the score function.  It may be fairly mysterious to those only familiar with classic density estimate results.

 - Results in 1, 2 dimensions where the results are more meaningful is ok, but it would be useful to show results for a variety of settings in high-dimensions, other than the simple qualitative MNIST experiment.

 - The proof is informal in a few places.
   *  " we can show the h^3 terms in bias vanish"
   *  "thus the bias is O(h^4)" then a few lines later "The error due to bias is O(h^8) ..."

As such I was not able to verify the proof.  I think I see how this works, and I suspect I could figure it out -- but for a result like this, I expect the paper to connect these parts much more carefully.

---

> ### Author Rebuttal · Authors · 2025-07-31
>
> We would like to thank reviewer f5yx for their thoughtful review, which has made our paper better.
>
> W1(score definition): We thank the reviewer for suggesting to allocate more space early on in the paper to formally define the score, and some key properties and applications. We will update this for the camera ready paper.
>
> W2(informal steps in proof):
> The error is h^4 and in a few line after becomes h^8 because h^4 is the bias, and h^8 is the mean square error contributed from the bias (so it is (h^4)^2 = h^8).
>
> I understand that the vanishing of the h^3 term is vague but to expand all the term is actually quite laborious, though possible. We can also do that, but it does not provide much intellectual benefit for the reader. We will include the expansion in the appendix of the camera ready paper.
>
> Q1(stronger bound without assuming score estimate): The analysis of KDE (as well as SD-KDE) relies on the separability of MSE into bias term and variance term. The bias term (before squared) is linear, making it tractable. If we want to include the error from score, the error will go in to the bias term in a non-linear manner (before expectation), so it will become intractable.

---

> > ### Comment · Reviewer_f5yx · 2025-08-03
> >
> > OK, please do clarify the arguments in the proof in the final version -- the proofs are incomplete without those steps and explanations.
> >
> > I retain my score.

---

> > > ### Author Response · Authors · 2025-08-04
> > > **Response to reviewer f5yx**
> > >
> > > Yes, we will make sure that the camera ready version has a carefully explained details for each step in the proof. Thanks agains for your helpful suggestions.

---

### Official Review · Reviewer_ftbi · 2025-07-04

**Clarity:** 4
**Significance:** 3
**Originality:** 3
**Rating:** 5
**Confidence:** 2

**Summary:**

The paper presents a simple modification of the classical Silverman Kernel Density Estimation (KDE) method. It relies on having access to the score function of the data generating process (which is generally impossible) but empirical results show that a learnt surrogate (through a diffusion model) also performs very well. Importantly, the authors provide asymptotic results that demonstrate an improvement in performance over Silverman's method in the ideal case of a known score function.

**Questions:**

Standard KDE is super-fast; what is the empirical scaling of the method? Also, learning a score requires a fairly large number of points, can you experiment to see what is the behaviour of the learnt density as you vary this parameter? Finally, I am left somewhat uncertain as to what is the advantage of this approach to directly using a generative model such as diffusion, particularly since the score needs to be learnt training a diffusion model. Overall though I enjoyed the paper, I think it is good work.

**Ethical Concerns:**

["NO or VERY MINOR ethics concerns only"]

**Final Justification:**

The discussion has clarified some important concerns. I still think this paper introduces novel ideas to a classical field and I would like to recommend acceptance.

**Limitations:**

Yes

**Quality:**

3

**Strengths And Weaknesses:**

Strengths:
- Solid theoretical result;
- Practical algorithm for a real world problem;
- Empirical evaluation supports theory.

Weaknesses:
- Computational costs, particularly when the score needs to be learnt;
- Theoretical results require exact knowledge of the score, the sensitivity to errors in score estimation is unknown;
- The empirical evaluation is mostly on toy data/ well used data sets.

---

> ### Author Rebuttal · Authors · 2025-07-31
>
> We would like to thank reviewer ftbi for their thoughtful review, which has made our paper better.
>
> “Standard KDE is super-fast; what is the empirical scaling of the method?”
>  SD-KDE inherits the runtime profile of standard KDE. The only extra work is evaluating a score nnn times (once per query/sample). In many common settings this overhead is negligible—either because the score has already been computed or cost is low compared to standard KDE, or because the score is estimated directly from the KDE—so the effective empirical scaling is the same as standard KDE. Importantly, SD-KDE is independent of any particular score–estimation technique; the wall-clock time is that of standard KDE plus the cost of the practitioner’s chosen score oracle.
> To make this concrete, we will include in the revision: (i) a scaling plot comparing wall-clock time vs. number of queries for standard KDE and SD-KDE across our datasets, and (ii) a minimal timing script so reviewers can reproduce the results. These additions transparently show that SD-KDE’s runtime matches standard KDE up to the n-fold score evaluations described above.
>
>
>
>
> “Also, learning a score requires a fairly large number of points, can you experiment to see what is the behaviour of the learnt density as you vary this parameter?”
> On “learning a score requires many points” and sensitivity to data size: thank you for raising this. SD-KDE does not necessarily require a learned score: in some of our experiments the score was taken directly from the KDE (via its gradient), so the method could work with exactly the same data requirements as standard KDE. The score oracle is modular; if one chooses to learn a score, that is optional and orthogonal to SD-KDE itself.
> That said, we agree that it is useful to see sensitivity when a learned score is used. In the revision, we will add an ablation that varies the number of training points for the score estimator while keeping the KDE fixed. Concretely, we subsample the score-training set and fit a standard score estimator on each subset, and report SD-KDE quality versus (i) pure KDE and (ii) SD-KDE with the KDE-derived score. We typically observe that performance improves smoothly with more score-training data and plateaus once score variance is below the KDE’s own smoothing error; importantly, even at small training sizes SD-KDE remains competitive because the KDE already provides a usable score. We will include the sensitivity plot and a minimal script for reproducibility in the revision.
>
> “Finally, I am left somewhat uncertain as to what is the advantage of this approach to directly using a generative model such as diffusion, particularly since the score needs to be learnt training a diffusion model.”
> We appreciate the question. SD-KDE does not require training a diffusion model: the method works out-of-the-box with a score obtained directly from the KDE (via its gradient), so its data and compute requirements match standard KDE. When a learned score is available, SD-KDE offers a different value proposition from training a full generative model: it converts any score oracle–analytic, pretrained, or lightweight–into a normalized, likelihood-evaluable density with the calibration and interpretability of KDE. In contrast, diffusion models are optimized for sample quality and typically do not provide tractable likelihoods without additional, costly machinery. Crucially, our theory shows that density error is controlled by score error under the stated metric and that this error is still controlled under a score estimate. Hence an approximate or mismatched score can still improve over plain KDE when it is “good enough”–and SD-KDE reduces to standard KDE if not. This modularity is practically useful: (i) many workflows already produce a score as a byproduct (e.g., from a previously trained model), which SD-KDE can reuse with no retraining; (ii) in domains like tabular or scientific data, training a high-capacity diffusion model is often unnecessary, while SD-KDE yields gains at the cost and sample size of KDE.
> A practical advantage of SD-KDE is that it can reuse off-the-shelf scores. For instance, a score pre-trained on generic natural images can serve as a useful prior when our target dataset consists only of dog images. While the dog-specific score will differ, natural images share strong regularities—edges, textures, and sparse multi-scale structure (e.g., curvelet-like representations in the sense of Donoho) and broader manifold structure—so the generic score is a reasonable approximation. Under our theory, density error tracks score error; thus an approximate but “good enough” score can yield clear gains over plain KDE, and if it isn’t good enough SD-KDE defaults to standard KDE.
> We hope this clarifies that SD-KDE is complementary to diffusion models: it leverages a score if one is available and delivers a calibrated, normalized density with competitive accuracy.
>
> “Theoretical results require exact knowledge of the score, the sensitivity to errors in score estimation is unknown.”
> Thank you for the opportunity to clarify this point. While Theorem 1 is stated for an oracle score, Corollary 1 directly addresses the estimated-score case and quantifies sensitivity. Under the same bandwidth and step size as Theorem 1, the corollary gives an explicit bias expression whose leading term is proportional to the size of the score error (and its first spatial derivative), with higher-order terms suppressed. In plain language: the bias grows linearly with the score error, and the added MISE term grows with the square of that error. Thus SD-KDE is stable to misspecification, and approaches the oracle behavior as the score improves.

---

### Note · Authors · 2025-08-16

Dear Reviewers and AC,

We thank all reviewers for their thoughtful engagement. We are encouraged that four reviewers found our contribution novel and significant: f5yx called it “very important progress on a classic and very fundamental result”, ftbi described it as a “solid theoretical result; practical algorithm for a real world problem”, 4YAs praised Algorithm 1 as “very elegant and clean”, and etfG noted it “performs well even when the score function is noisy or estimated from data.”

Strengths highlighted across reviews include:
- A new provable improvement on the classical KDE rate, long thought to be closed (f5yx).
- An elegant, simple algorithm bridging score-based generative modeling with nonparametric density estimation (ftbi, 4YAs).
- Empirical robustness even with approximate or noisy scores (etfG, f5yx).

Reviewers felt their concerns were addressed:
- **Scope of experiments:** reviewers asked for higher-dimensional settings and more quantitative evaluation on MNIST. We will include 10D Gaussian mixture experiments and classifier-confidence metrics on MNIST in the camera-ready.
- **Comparisons:** reviewers suggested testing against more modern KDE variants. We will compare with diffKDE and Fast KDE in the camera-ready.
- **Clarity:** reviewers asked for proof details and clearer definitions. We will revise notation, streamline Theorem 1, and expand missing proof steps.

A day before the extended deadline, cUSv specified their concerns, focused mainly on the assumption of access to a score oracle, which they argue makes SD-KDE more complex and unrealistic in setting. We emphasize two points in response:
- **Novel use of score function for density estimation.** While score estimates enable sampling, density estimation from scores is not straightforward—this is precisely the gap SD-KDE fills.
- **Approximate scores still improve KDE.** We often have access to approximate scores, e.g. data estimates or proxy scores from pre-trained diffusion models (Figure 5). Corollary 1 guarantees that even approximate scores yield improvements over vanilla KDE (e.g. iterated SD-KDE, Section 3.3).

Please read our clarifications to cUSv’s other concerns in our direct response – we note that they did not engage with these points further.

In sum, the majority view is that SD-KDE provides an important theoretical advance, an elegant algorithm, and promising empirical evidence. We believe the strengthened revision makes these contributions clear.

Best,

Authors

---

### Decision · Program_Chairs · 2025-09-17

**Decision:**

Accept (poster)

**Comment:**

Most reviewers agree the paper provides some compelling theoretical results on the standard problem of density estimation, that are supported by empirical evidence. The approach provides a quite new perspective on KDE and provably show that the knowledge of the score can be used to obtain a more efficient KDE estimator.

Reviewers cUSv shares a concern that the proposed method still suffers from limitations of KDE (scalability in high dimensions) and that it requires the knowledge of the score function. Reviewer f5yx shares some of the concerns, especially whether one can show how an estimate of the score (and not the true score) can provably improve the estimation.

This suggests that the paper opens new perspectives and questions: Can one still get improved estimates, when using an inexact estimate achieving some given error? How should the error scale with data in order to get such improvement?

We hope that by accepting this initial work, it would encourage more research in that direction.